# Implementing public health emergency operations centres according to an international framework in Ethiopia, Nigeria, and Senegal: Best practices and achievements, 2021

Senait Tekeste Fekadu[1]*, Jian Li[2], Yangmu Huang[3], Wessam Mankoula[4], Womi Eteng[4], Ali Abdullah[5], Virgil Lokossou[6], Chuck Wilton Menchion[7], Emily Rosenfeld[7], Zhuo Li[3], Jiyan Ma[3], Chunshan Zhao[3], Zhi Jie[3], Abrham Lilay Gebrewahid[1], Ishata Conteh[1], Aschalew Abayneh[8], Zewdu Assefa[8], Shambel Habebe[8], Everistus Aniaku[9], Emmanuel I. Benyeogor[9], John Oladejo[9], Alioune Badara Li[10], Abdoulaye Bousso[10], Ibrahima Sonko[10], Alle Baba Dieng[10], Fiona Braka[1], Abdou Salam Gueye[1], Ihekweazu Chikwe[2]

1 World Health Organization Regional Office for Africa, Brazzaville, Republic of Congo, 2 World Health Organization, Headquarters, Geneva, Switzerland, 3 Peking University, Beijing, China, 4 Africa Centres for Disease Control and Prevention, Division of Emergency Preparedness and Response, Addis Ababa, Ethiopia, 5 World Health Organization Regional Office for Eastern Mediterranean, Cairo, Egypt, 6 West African Health Organization, Abuja, Nigeria, 7 US Centres for Disease Control and Prevention, Atlanta, Georgia, United States of America, 8 Ethiopia Public Health Emergency Operations Centre, Addis Ababa, Ethiopia, 9 Nigeria Public Health Emergency Operations Centre, Abuja, Nigeria, 10 Senegal Public Health Emergency Operations Centre, Dakar, Senegal

* tekestes@who.int

## Abstract

A public health emergency operations centre (PHEOC) is a hub for effective coordination of information and resources. Countries have established PHEOCs as part of the effort to strengthen their emergency management capabilities. However, there is limited documented evidence of best practices in PHEOC implementation in accordance with the world health organization PHEOC framework. A survey was conducted to document best practices and experiences in implementing PHEOCs in Ethiopia, Nigeria, and Senegal. Institutional and individual-level data on PHEOC implementation were collected between 6 December 2020 and 25 March 2021 through structured survey administered to regular and surge staff. An accuracy rate above 70% was considered adequate for knowledge of PHEOC operations, while individual emergency management competency was classified as adequate if the score was at least 3.5. Ethiopia and Nigeria established PHEOCs with legislation, while Senegal issued a ministerial directive. Senegal's PHEOC had better facilities, whereas the PHEOC in Ethiopia and Nigeria had facilities with limited capacity to meet surge demands. Plans and procedures were in place to guide the PHEOC operations and an incident management system to coordinate responses. The PHEOCs had skilled staff but were inadequate for large-scale emergencies. Senegal and Nigeria had well-established information systems to support PHEOC

**Data availability statement:** All relevant data are within the paper and its Supporting information files.

**Funding:** The author(s) received no specific funding for this work.

**Competing interests:** The authors have declared that no competing interests exist.

operations. All activated their PHEOC before detecting a coronavirus disease 2019 (COVID-19) case to coordinate readiness efforts. The staff was knowledgeable in PHEOC operations, with demonstrated capacity in emergency management. There were constraints in developing hazard-specific plans and funding. Overall, the three countries improved multisectoral coordination and emergency response capacity. Key lessons highlight the importance of strong legal mandates, standardized plans and procedures, trained routine and surge workforce, and functional information systems. These experiences provide valuable insights for other countries, although further efforts are needed to strengthen PHEOC systems.

## Introduction

The International Health Regulations (IHR 2005) require countries to build their capacity to prevent, detect, and respond to public health emergencies (PHEs). Specifically, IHR (2005) requires that States Parties develop, strengthen, and maintain their capacity to respond promptly and effectively to public health risks and emergencies. A public health emergency operations centre (PHEOC) is a hub for the effective coordination of information and resources during management of PHEs and meets IHR (2005) requirements [1–3]. A PHEOC supports multisectoral information and resource coordination by establishing a clear structure at strategic, operational, and tactical levels [3–5]. PHEOC uses an incident management system (IMS) defined within plans and procedures as the best practice model to coordinate emergency response operations.

To promote best practices and standards to support the implementation of PHEOCs, the WHO published "Framework for Public Health Emergency Operations Centres" in 2015. In addition, WHO Regional Office for Africa published a handbook for PHEOC operations and management and a Guide for Development of a legal instrument for the establishment and operationalization of a PHEOC [4,6,7]. Meanwhile, the must-read checklist and implementation guidance are beneficial for the member states to establish public health emergency preparedness and response strategies based on their specific geopolitical and socioeconomic status [4].

The 2014–2016 Ebola virus disease (EVD) outbreak in West Africa showed the need to strengthen health emergency preparedness and response systems. With limited technical, financial, and human resources, health emergencies like EVD could have disastrous consequences on the continent, have already strained health systems, and could quickly turn into a social and economic disaster. In August 2016, African governments assembled at the 66th session of the Regional Committee for Africa with a set of political commitments by passing the resolution to adopt the regional strategy for health security and emergencies 2016–2020 [8]. The strategy urges that "At least 80% of Member States have a PHEOC meeting the minimum common requirement by 2020 [9]." Since then, Member States in Africa have developed a PHEOC based on lessons learnt from responses to different public health events. However, there is limited evidence on implementation of best practices defined in the

PHEOC framework in Africa [10]. Despite the availability of normative guidance, there is limited empirical consensus on how PHEOC best practices are operationalised across Member Stated in Africa. It is key that countries implement emergency management best practices adapting to their context and aligning with their policy, plans, and procedures.

Ethiopia, Nigeria, and Senegal are among the countries that established PHEOCs to support response coordination to specific disease outbreaks as in Nigeria which was established for polio outbreak response in 2014. All these countries established their PHEOCs before the COVID-19 pandemic and played role in leading multi-sectoral coordination during the management of the COVID-19. Senegal's PHEOC has been directly administered by the Ministry of Health (MoH) since December 1st, 2014, and has been under the MoH's General Directorate of Public Health since March 2018. Nigeria's PHEOC is situated in the Nigeria Centres for Disease Control and Prevention (NCDC). In Ethiopia, the PHEOC is under the leadership of the Ethiopian Public Health Institute (EPHI) and established in 2017. These countries were selected because of their interest to share their experiences including identifying best practices in establishing PHEOCs to improve emergency management. Besides, experts from their PHEOCs were instrumental to support the implementation of PHEOC in other countries and were interested to share their experiences in PHEOC implementation. Furthermore, the Joint External Evaluation (JEE) conducted in the three countries indicated critical gaps in establishing effective coordination for the management of PHEs.

The purpose of this article is to document and compare the best practices and implementation experiences of PHEOCs in Ethiopia, Nigeria, and Senegal to improve their public health emergency management (PHEM) capabilities and assess how these align with WHO PHEOC Framework. –. It could also be used as a reference for other countries to adapt the best practices to further improve their PHEOCs.

## Methods

### Ethics statement

The data collected pertains solely to policies, plans, procedures and systems, without involving human subjects. Participants from the three countries agreed verbally after being informed about the purpose and objectives, and by delivering their replies, they agreed to participate. The data collected for this study was anonymised and no personally identifiable information included.

### Data collection

Two questionnaires were employed to assess the implementation of the four core components of PHEOC and emergency management competencies at the institutional and individual levels (S1 Data). Data were collected between December 6, 2020, and March 25, 2021. Independent variables considered included country of work, position in the institution, functional role, education level, gender, age, sources of training, and satisfaction with the training programme. Dependent variables including knowledge and concepts, skills, and core competencies required for PHEs, as well as the diversity of job functions of employees were considered. The survey was administered in English through Survey Monkey, an open-source online survey application, with informed consent obtained through inclusion in the survey tool and data anonymized. The WHO AFRO contacted the PHEOC focal points of the three countries through respective WHO country offices and PHEOC managers, incident managers and surge staff working under the different thematic areas within the PHEOC operations were invited to complete the survey.

**Institutional level data.** For the institutional survey, PHEOC managers were asked to complete 99 closed and open-ended questions. Of these, 43 questions were regarding PHEOC's initial development, characteristics, and setting, 29 questions on PHEOC legal frameworks, plans and procedures and 29 questions related to COVID-19 preparedness and response. Three PHEOC managers and two incident managers (IM) were virtually interviewed to verify and clear the data collected through the survey tool. Questions were formulated based on the feedback reflected in the data and most

of the questions were open-ended to allow an in-depth understanding of the PHEOC implementation and best practices according to an international PHEOC framework.

**Individual level data.** A total of 76 staff (Ethiopia: 36, Nigeria, 15 and Senegal: 22) were invited and all completed the individual survey, which included 124 questions. The survey was divided into three parts. Part A about knowledge and concepts of PHEOC operations mainly through true-false and multiple-choice questions. Part B covered the skills and abilities needed for daily PHEOC operations and staff conducted self-rated on the following PHEOC function-specific roles: 1) policy, 2) management, 3) operations, 4) planning, 5) communication, 6) logistics, 7) finance and administration [4]. Part C addressed the core competencies required of response personnel to ensure effective management of PHEs including leadership capacity, capacity in understanding emergency management principles and doctrines, integrating crisis and emergency risk communication to an incident, training facilitation and evaluation.

## Data analysis

Thematic analysis was used to summarise the institutional-level key findings on the implementation of the four core components of PHEOC. At the individual level, an accuracy rate higher than 70% was considered adequate for the knowledge, skills and understanding of PHEOC emergency management operations. All self-rated items were measured on a 5-point Likert scale (1 = very limited, 5 = very well); for analytical purposes, an average score of ≥3.5 was operationally defined as "adequate" competency, as it reflects performance consistently above the neutral midpoint (3.0) and approximates ≥70% of the maximum attainable score, a convention commonly applied in public health capacity assessments to facilitate interpretation rather than as a psychometrically validated benchmark. Scores below 3.5 were classified as limited. Multivariate logistic regression analysis was used to identify factors influencing individual response capacity, with statistical significance set at $p < 0.05$.

## Results

### Institutional-level capacities

**Legal instruments, plans and procedures.** The legal instrument to define PHEOC mandates and provide authority for response operations has been established in all three countries, though in different ways. PHEOCs were authorized by legislation in Ethiopia and Nigeria, and by an executive directive from the health minister in Senegal [11]. The legal instrument provided mandates to lead the management of PHEs, defined the multisectoral coordination system with other sectors, and mobilization of funding.

Ethiopia had a permanent multi-sectoral policy group called the public health emergency management task force. Senegal and Nigeria had an ad hoc policy group when necessary. The task forces were crucial in providing strategic direction, leadership, and availed resources for PHEOC operations.

The plans and procedures including the concept of operations (CONOPs) of the PHEOCs in the three countries were designed to make the response processes routine, systematic, and predictable. When pre-established criteria are met, the PHEOC in these countries would be activated after conducting risk assessments. Plans and procedures including PHEOC operations and management handbooks, the multi-hazard, and hazard-specific preparedness and response plans helped to ensure clear guidance, coordination and communication among the various actors. IMS structure was established with a set of standard operation procedures (SOPs) and protocols, including SOPs for communication and information sharing, and management of rapid response teams (RRTs). All three countries had procedures for credentialing and permission to response personnel who support response operations.

**Human resources, training, and exercise.** It is important to train PHEOC regular and surge staff on IMS and role-specific functions prior to the occurrence of PHE and maintain an updated roster to facilitate staffing of various IMS functions to support effective emergency management [12,13]. The three countries had access to a variety of regional and international dedicated training and exercise programmes to regularly train their surge staff (Table 1). Nigeria and

**Table 1. Capacity building opportunities for response personnel – Ethiopia, Nigeria, and Senegal, 2021.**

| | Senegal | Ethiopia | Nigeria |
|---|---|---|---|
| National PHEOC training and exercise | √ | √ | √ |
| WHO Global EOC-NET | √ | √ | √ |
| WHO Regional PHEOC-NET (AFR) | √ | √ | √ |
| Regional PHEOC simulation exercise | √ | √ | √ |
| Global Pandemic Response Simulation Exercise | √ | / | √ |
| Regional PHEOC training | √ | √ | √ |
| Staff trained as trainers and deployed to support other countries. | √ | / | √ |
| GOARN Training Workshop | / | √ | √ |
| Rapid response team training | √ | √ | √ |
| Other programmes (specify) | Disaster Management, Bournemouth University and WAHO/US CDC online PHEM training | | WAHO/US CDC online PHEM training |
| Most benefited from | Regional PHEOC training | Regional PHEOC SimEx | WHO Global EOC-NET |

Senegal gained additional access to sub-regional training and exercise programs. All three countries have established a database of trained experts in PHEOC operations and IMS, as well as procedures for mobilizing them to fill various IMS functions when their PHEOC is activated to coordinate PHE management. The sources for the surge staff were different government ministries and sectors including regional and international partner organizations. In comparison, Nigeria experienced a shortage of surge staff, which was due in part to a lack of well-trained personnel and funding. The Nigeria PHEOC implemented a flexible plan that allowed response personnel to work in multiple technical areas at the same time within the PHEOC and implementing an updated surge staff recruitment plan helped to address the gap.

**Data and information.** Information is the lifeblood of a PHEOC and three types of data (event-specific, event management information, and context) are required in the PHEOC to facilitate the decision-making process and systems need to be established to ensure the effective flow of information to the PHEOC [14,15]. The three countries have established well-developed information systems to support PHEOC functions. Nigeria has used a surveillance outbreak response management application system (SORMAS) platform to collect and manage data on disease outbreaks [16]. Both Senegal and Ethiopia used a real-time information system down to the community/district level for disease monitoring – the District Health Information System 2 (DHIS2), to collect and analyse information and display interactive information through digital signage technology. Besides, Senegal used the Minfo Sante application that is used for event-based surveillance, contact tracing, ambulance management, meeting management, communication, and notification among actors. Ethiopia PHEM staff had limited capacity in determining the data set needed and analysing to facilitate decision-making process.

**Technology and physical infrastructure.** A PHEOC needs to acquire the technologies such as computers, phones, TV plasma screens, internet connectivity, and radio systems to facilitate telecommunications, and information management during response coordination. The PHEOC in Senegal takes a space of 1,200 square meters up and is housed in an independent building and is adequate to accommodate more surge staff during the emergency response. Whereas Nigeria PHEOC (12 sq. meters) and Ethiopia PHEOC (45 sq. meters) are situated in an existing facility and limited space capacity to accommodate surge staff mobilized to support response operations. During the COVID-19 pandemic that required working spaces to accommodate more surge staff, Nigeria rapidly developed a plan to respond to physical space shortage and manage the number of participants in in-person meetings by promoting online offices,

holding online meetings, and 30 sub-national PHEOCs within its PHEOC network. Similarly, Ethiopia repurposed syndicate rooms that are routinely used for training activities. Various information technology infrastructures have been established in all three countries. In comparison, Senegal, and Nigeria had established relatively adequate office equipment, video, and teleconference equipment.

### Individual-level capacities

**Self-reported knowledge and concepts of PHEOC operations.** The average accuracy rate of the staff on PHEOC operational knowledge and concepts in the three countries was higher than 70%. Staff in Nigeria had the highest level of knowledge and concepts of PHEOC operation with a test accuracy rate of 75%, as well as the best understanding of the "concepts of operations of the PHEOC", which is a key requirement for coordination between different sectors at strategic, operational, and tactical levels. Staff in Senegal had a relatively limited grasp of the elements of the emergency management plan ($\beta$=-2.823, $p<0.05$). Also, among the three countries, it is worth mentioning that compared with staff aged 25–34, staff aged 35–44 had a significantly better understanding of the elements of emergency management plans ($\beta$=-1.8129, $p<0.05$). Overall, staff in Ethiopia, Nigeria, and Senegal respectively rated themselves scores of 3.67, 3.82, and 3.59 out of 5 in the knowledge and concepts of "advanced emergency management and PHEOC operations". Analysis results on the accuracy rate on knowledge and concepts and the variables used in the individual capacity analysis are presented in Table 2.

**Self-reported PHEOC function-specific roles.** As shown in Table 2, staff in the three countries performed best in a communication role, while there was a need for improvement in finance and administrative functions. Compared with staff in Ethiopian, staff in Nigerian and Senegal performed relatively limited in operation (Nigeria: $\beta$=-0.1299, $p<0.05$; Senegal: $\beta$=-0.1412, $p<0.05$) and planning functions (Nigeria: $\beta$=-0.1611, $p<0.05$; Senegal: $\beta$=-0.1659, $p<0.05$). The overall individual capacity of Senegal staff was scored lower ($\beta$=-0.1036, $p<0.05$) than that of the other two countries.

Personnel that received limited training self-rated as having lower capacities in planning ($\beta$=-0.1849, $p<0.05$) and less job diversity ($\beta$=-0.1777, $p<0.05$) as compared to personnel that have received adequate training on PHEOC operations. Personnel who have not received any formal training related to PHEOC rated themselves as having lower abilities in functional roles related to policy ($\beta$=-0.1789, $p<0.05$), management ($\beta$=-0.1527, $p<0.05$), and planning ($\beta$=-0.2082, $p<0.05$). Staff whose roles involve PHEOC logistics had limited abilities in carrying out management ($\beta$=-0.4846, $p<0.05$), operations ($\beta$=-0.3118, $p<0.05$), and planning ($\beta$=-0.4092, $p<0.05$) related functions.

**PHEM core competencies.** As shown in Table 3 Senegal employees had an average score of less than 3.5 in evaluation, emergency management, and emergency management communication. In addition, compared with Ethiopian employees, Nigeria ($\beta$=-0.8726, $p<0.05$) and Senegal employees ($\beta$=-0.9824, $p<0.05$) had a significantly lower core competency in public health emergency management. The higher the education level of the staff, the more confident they are at work

**Table 2. Self-rating of PHEOC-function-specific knowledge and abilities – Ethiopia, Nigeria, and Senegal, 2021.**

| PHEOC functions | Ethiopia | Nigeria | Senegal | Average |
|---|---|---|---|---|
| Policy | 3.98 | 3.77 | 3.85 | 3.92 |
| Management | 4.15 | 4.64 | 4.03 | 4.00 |
| Operations | 4.20 | 3.83 | 3.88 | 4.09 |
| Planning | 4.09 | 3.37 | 3.68 | 3.85 |
| Communication | 4.33 | 3.94 | 3.74 | 4.14 |
| Logistics | 3.92 | 3.46 | 3.68 | 3.82 |
| Admin and Finance | 3.53 | 3.57 | 3.52 | 3.53 |

**Table 3. Self-rating average scores of core competencies required of response personnel – Ethiopia, Nigeria, and Senegal, 2021.**

| Core competency | Ethiopia | Nigeria | Senegal | P-value |
|---|---|---|---|---|
| Leadership capacity | 3.51 | 3.14 | 3.51 | 0.615 |
| Capacity in understanding and applying overall emergency management framework | 3.53 | 3.44 | 3.55 | 0.952 |
| Capacity in Emergency management | 3.59 | 3.33 | 3.42 | 0.667 |
| Capacity in Emergency management communication | 3.70 | 2.57 | 3.09 | 0.019* |
| Capacity in Partnership and Collaboration | 3.82 | 3.25 | 4.06 | 0.197 |
| Capacity in Training Development and Facilitation | 3.72 | 2.95 | 3.58 | 0.200 |
| Capacity in Evaluation | 3.69 | 2.65 | 3.29 | 0.017* |
| Overall core competencies | 3.65 | 3.05 | 3.44 | 0.194 |

($\beta = 0.894$, $p < 0.05$). Compared with individuals who had received training on understanding and operating PHEOCs and had sufficient knowledge and skills, personnel who had received a small amount of training but do not meet the job requirements ($\beta = -0.8062$, $p < 0.05$) and personnel who had not received any kind of formal training related to PHEOC ($\beta = -1.1111$ $p < 0.05$) had a lower capacity in understanding and applying overall emergency management framework.

### Self-reported readiness and response to COVID-19

All three countries were relatively well prepared and responded quickly in the early stages of the COVID-19 outbreak. They have established a relatively adequate system that was activated for readiness and response efforts. The PHEOCs in three countries formulated readiness plans and incident action plans before the index case was identified in the country. The PHEOC in these three countries conducted rapid risk assessments and determined the level of risk and guided the shift from one to another PHEOC mode of operation in line with the procedures that were in place before the occurrence of COVID-19 as depicted in Fig 1 [17]. Both Nigeria and Senegal activated their PHEOCs right after the index case in the country was identified, while Ethiopia activated its PHEOC to a moderate level long before the first case for readiness activities. PHEOC in Ethiopia determined their risk as high in the very early days considering their airports and ports' health services as one of the most important pivots among Africa and other continents, and when the index case was identified, they increased the activation level to the highest level to devote full efforts in response to the outbreak.

Immediately after the initial activation of the PHEOCs, all three countries established their Incident Management System (IMS) to strengthen multi-sectoral coordination of information, resource, and staff workforce which proves crucially important in emergency responses. In Ethiopia, the PHEOC shared highlights in daily situation report and weekly bulletin to relevant stakeholders for informed decision-making and better coordination. PHEOC of Nigeria immediately gathered the nine functional pillars (coordination, surveillance and epidemiology, case management, laboratory, points of entry (POE), infection prevention and control (IPC), risk communication, logistics, and research) together. Biweekly meetings were held in the IMS for the presidential task force to discuss the priorities and challenges and find out the best practices across states in Nigeria. The PHEOC in Senegal held daily meetings to discuss the data and information obtained from the regional level and promptly adjust the action plans in consideration of the situation.

Overall, the PHEOCs of all three countries have adopted similar interventions based on their plans and procedures for early-stage of preparation and response to COVID-19. Institutional managers in the three countries all stated that they had taken corresponding and effective actions at different stages of the outbreaks.

### Discussion

This study aimed to document the implementation of the PHEOC according to recommended standards in the international PHEOC framework in Ethiopia, Nigeria, and Senegal PHEOCs. The results showed that PHEOC plays a critical

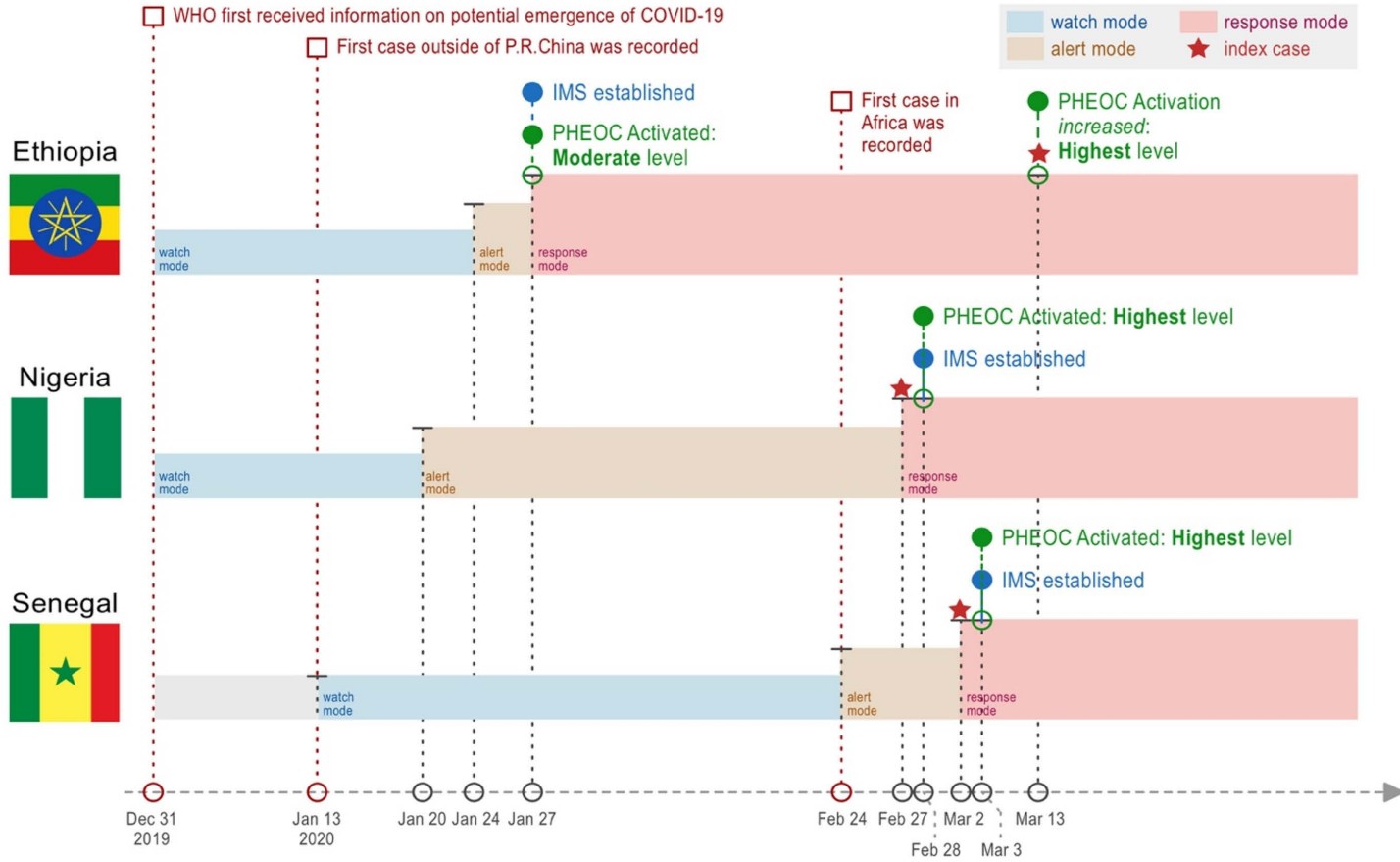

**Fig 1. Timeline of early-stage response to COVID-19 – Ethiopia, Nigeria, and Senegal, 2021.**

role in preparing for and responding to public health emergencies. Overall, the development and operationalization of PHEOCs have proven successful in emergency management in these three countries. Our results showed that implementing the four PHEOC key components including legal authority, plans, and procedure; data and information for action; infrastructure and technology, and skilled human resources have played critical roles in effective PHE management including COVID-19. Having this system in place proves to be effective in improving their coordination of resources and information, as well as coordination among multiple sectors. At the early stage of the COVID-19 pandemic, these governments coordinated internal and external resources made relatively appropriate preparations and formulated relatively detailed incident action plans based on the standardized operating procedures and experience accumulated in dealing with previous public health emergencies through the PHEOC system and IMS structures. Findings from Nigeria align with this study, documenting that establishment of national and sub-national PHEOCs has strengthened coordination of outbreak preparedness and response activities [18]. However, there were still some challenges that the PHEOCs faced, and it is suggested to make further improvements in terms of hazard-specific plans, information systems, infrastructure and technology, and human resources.

Adequate legal authority and policy formulation enable solving challenges that threaten the effective operations of PHEOC [19]. Legal authority defines a PHEOC's mandate, clearly outlines its roles and responsibilities, supports the decision-making processes, its coordination mechanisms with national and international disaster management resources, response activation and deactivations parameters, policy group members and roles, and funding mechanism for the

operations of a PHEOC, among other roles and responsibilities [4]. Our results indicate that PHEOCs in Senegal, Nigeria, and Ethiopia endorsed a legal authority to ensure the engagement of various sectors as the establishment of appropriate legal authority helps to involve relevant stakeholders working in public health emergencies [19]. It is worth noting that Senegal establishes its legal authority through executive directives with effective management and coordination. Thus, the national government could coordinate relevant stakeholders from different departments, sectors, and organizations to conduct its legal authority in line with the national policies and plans. Policy groups also play a key role in the operation of the PHEOC, particularly in providing strategic guidance, decision-making and endorsing external resource requests from regional and international donors and partners [7]. It was found that Ethiopia had a standing policy group however Nigeria and Senegal had ad hoc policy groups.

The PHEOC plans and procedures are the primary resources for PHEOC staff, which describe the structure, functions, concept of operations, and procedures for operating a PHEOC. Various plans and procedures provided a foundation for the implementation of the ultimate mission, including PHEOC plans, multi-hazard plans, incident or hazard-specific response and management plans, incident action plans, and other functional plans [9]. Described in plans and procedures, IMS is regarded as a recommended response model. It is worth noting that the operational pillars in the IMS structure are not always the same but are designed according to different situations [ 20,21]. National emergency management policies and plans should be considered and aligned in making PHEOC plans and procedures. According to our study, all three countries have developed PHEOC plans and established functional IMS structures, and when the PHEOC is activated and can develop and revise a series of incident action plans over the defined operational periods as emergency status changes.

Across the three countries, Joint External Evaluation (JEE) missions conducted between 2016 and 2017 identified low to moderate operational capacity, with Ethiopia scoring 1–2 across key response indicators, Nigeria scoring 2–3, and Senegal demonstrating relatively stronger performance with scores of 2–3 for response activation, PHEOC plans and procedures, emergency operations programmes, and case management (JEE scoring scale: 1 = no capacity; 2 = limited capacity; 3 = developed capacity; 4 = demonstrated capacity; 5 = sustainable capacity). These indicators correspond primarily to JEE domains R1–R4, which assess national legislation and financing, IHR coordination, preparedness, and emergency response operations. In contrast, the present study demonstrates significant progress across all three settings through the establishment and operationalization of national PHEOCs, earlier activation for readiness and response, routine use of Incident Management Systems, and strengthened legal authority, policies, plans, and procedures to standardize preparedness, response, and recovery. These advances indicate progress toward "developed" and, in some domains, "demonstrated" capacity under the IHR (2005) monitoring framework [22–24]. However, our results reflect the shortages of hazard-specific response and management plans in three countries, and additional coordination and communications with sub-national regional levels should be made within the plans and procedures.

Daily PHEOC operations depend highly on physical infrastructure and information communication technology. Our study elucidates that all three countries possessed different capacities for developing infrastructure and facilities. However, there is still room for improvement in the aspect of physical infrastructure. In addition, the three countries should emphasize backup (physical site and technological) PHEOC, which could be activated if the primary site is not functional to ensure continuity of business. Besides, Ethiopia's technology infrastructure was relatively outdated, and improvements could be made in computer workstations with the necessary software applications, test phones, interoperable radio communication equipment, as well as data processing and analysis capabilities. The former is to meet the staff's workspace needs and ensure their safety, and the latter is to increase the availability, accessibility, quality, timeliness, and usefulness of emergency operations information for public health action [9].

Our study revealed that the system that the three countries have put in place assisted them to receive timely information from the community level, early case detection, analysis, and integration to inform decisions. There are however limited operational information management tools in the three countries including incident recording, response operations

tracking, resource management and tracking of partners' activities which are key for emergency response information management.

Our findings suggested that regular training and the conduct of simulation exercises significantly impact improving human resources and the development of core competencies. The three countries have carried out various training programmes and the attitudes towards receiving PHEOC training in various countries are very positive. However, about half of the employees from the three countries stated that the knowledge and skills involved in the training were still insufficient to meet the needs of the job, and more targeted training is needed to meet specific needs. Training in risk/hazard vulnerability analysis, IMS in action, planning, information, and communication technology especially needs to be further strengthened. This is similar to the lesson learnt from the Nigeria study, where trained PHEOC staff are frequently posted and reassigned to different ministries, departments, and agencies. This leads to a high level of PHEOC staff turnover and the necessity to continue training newly assigned personnel, interfering with public health response actions [18]. In addition, financial management is a critical factor in the effective operation of an organization [25]. Lack of basic emergency financial management knowledge and skills is an obstacle for employees to fully utilize their work abilities and achieve best practices [26]. The performance of the three institutions in financial management needs to be improved. It's critical to make sure that all of the PHEOC's main tasks, as defined by the WHO's Framework for a PHEOC, are context-aware as one of the lessons learnt was the likelihood of financial and administrative tasks left out, hurt PHEOC operations [18]. In addition to the institutions' basic training on PHEOC knowledge, more targeted training should be considered. It can be concluded that the PHEOCs need to establish an environment of continuous learning among all staff and formulate sustainable training and exercise plans/programmes according to their actual needs to regularly test and improve systems.

## Strengths and limitations

This pilot study highlights the best practices of PHEOCs in three countries, which gives an insight into PHEOC operations in Africa. Notably, the study was conducted amid the COVID-19 pandemic and objectively reflected the reactions of countries to public health emergencies. After two years of the COVID-19 pandemic, additional actions have been taken, and employees' knowledge, skills, and attitudes toward PHEOCs may also be improved. Future research should consider examining impact of PHEOCs in improving the emergency management system before and after their implementation. Overall, this is the first structured research to evaluate the best practice of PHEOCs in Africa based on the WHO PHEOC framework, which laid the adequate groundwork and developed a valuable evaluation model that could be highlighted and referenced.

## Conclusion

This study demonstrates that operationalizing PHEOCs in alignment with WHO PHEOC framework strengthens national capabilities for preparedness, response, and recovery from public health emergencies. Based on the experiences from Ethiopia, Nigeria, and Senegal, several key lessons learned that are applicable to other countries seeking to establish or strengthen PHEOCs.

First, establishing a clear legal mandate for PHEOC, whether through legislation or executive direction, is essential to enable clearly defined responsibilities, rapid activation, multi-sectoral coordination, and resource mobilization during emergencies.

Second, pre-defined policies, plans and procedures, and IMS are critical for ensuring predictable, systematic, structures and scalable response operations, particularly during large-scale or prolonged emergencies such as COVID-19.

Third, sustained investment in a trained and deployable routine and surge workforce supported by regular training and simulation exercise, is crucial to maintain functional readiness and mitigate the efforts of staff turnover.

Fourth, interoperable and real time information systems that integrate surveillance, operational and contextual data significantly enhance situational awareness and evidence informed decision making within PHEOC preparedness, response operations and recovery.

Lastly, adequate physical infrastructure and enabling technologies are required to support surge capacity and continuity of operations, particularly, during high demand period. Though the three countries have met the minimum requirement for a fully functioning PHEOC, there is still room for improvement to further strengthen their emergency management capacities in terms of allocating funding from the government to sustain the PHEOC, further developing surge staff capacities, and establish training and exercise programme to regularly improve systems. Nigeria and Senegal need to establish a standing muti-sectoral policy group supported by a legal framework to further enhance multi-sectoral coordination, collaboration, and information sharing. Furthermore, three countries need to develop an information system to manage incident-related operational and contextual information.

This study provides an opportunity to document the three countries' best practices, demonstrating that implementing the PHEOC standards enhances emergency management capacities. It also assisted in identifying gaps and lessons learnt and urges other countries to learn to strengthen PHEOC capacity according to the WHO PHEOC standards.

## Supporting information

**S1 Text. Survey tool used to assess PHEOC implementation.**
(PDF)

**S2 Text. Survey tool used to assess PHEOC staff competency.**
(PDF)

**S1 Data. Data.**
(XLSX)

## Author contributions

**Conceptualization:** Senait Tekeste Fekadu, Jian Li, Yangmu Huang.

**Data curation:** Senait Tekeste Fekadu, Jian Li, Yangmu Huang, Zhuo Li, Jiyan Ma, Chunshan Zhao, Aschalew Abayneh, Zewdu Assefa, Shambel Habebe, Everistus Aniaku, Emmanuel I Benyeogor, John Oladejo, Alioune Badara Li, Abdoulaye Bousso, Ibrahima Sonko, Alle Baba Dieng.

**Formal analysis:** Senait Tekeste Fekadu, Jian Li, Yangmu Huang, Zhuo Li, Jiyan Ma, Chunshan Zhao, Zhi Jie.

**Funding acquisition:** Senait Tekeste Fekadu.

**Investigation:** Senait Tekeste Fekadu, Jian Li, Yangmu Huang, Jiyan Ma, Chunshan Zhao.

**Methodology:** Senait Tekeste Fekadu, Jian Li, Yangmu Huang.

**Project administration:** Senait Tekeste Fekadu, Jian Li, Yangmu Huang.

**Resources:** Senait Tekeste Fekadu, Jiyan Ma.

**Software:** Yangmu Huang, Jiyan Ma.

**Supervision:** Senait Tekeste Fekadu.

**Validation:** Senait Tekeste Fekadu, Jian Li, Yangmu Huang, Jiyan Ma, Chunshan Zhao, Zhi Jie, Ishata Conte, Aschalew Abayneh, Zewdu Assefa, Shambel Habebe, Everistus Aniaku, Emmanuel I Benyeogor, Alle Baba Dieng.

**Visualization:** Senait Tekeste Fekadu, Yangmu Huang, Chunshan Zhao, Zhi Jie.

**Writing – original draft:** Senait Tekeste Fekadu, Jian Li, Yangmu Huang, Zhuo Li, Jiyan Ma, Chunshan Zhao, Zhi Jie, Alioune Badara Li, Ibrahima Sonko.

**Writing – review & editing:** Senait Tekeste Fekadu, Wessam Mankoula, Womi Eteng, Ali Abdullah, Virgil Lokossou, Chuck Wilton Menchio, Emily Rosenfeld, Zhuo Li, Zhi Jie, Abrham Lilay Gebrewahid, Ishata Conte, Aschalew Abayneh,

Zewdu Assefa, Shambel Habebe, Everistus Aniaku, Emmanuel I Benyeogor, John Oladejo, Alioune Badara Li, Abdoulaye Bousso, Ibrahima Sonko, Alle Baba Dieng, Fiona Braka, Abdou Salam Gueye, Ihekweazu Chikwe.

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
