## [Decision Letter · Decision Letter 0]

24 Sep 2025

PGPH-D-25-01752

Implementing public health emergency operations centres according to an international framework in Ethiopia, Nigeria, and Senegal: Best practices and achievements, 2021

Dear Dr. Fekadu,

Thank you for submitting your manuscript to PLOS Global Public Health. After careful consideration, we feel that it has merit but does not fully meet PLOS Global Public Health’s publication criteria as it currently stands. Therefore, we invite you to submit a revised version of the manuscript that addresses the points raised during the review process.

Please ensure that all comments from both reviewers are fully addressed. In particular, as both reviewers noted, the methods section requires clarification. Additionally, the discussion section should be strengthened by elaborating on how these findings compare with other studies and by highlighting their broader implications.

We look forward to receiving your revised manuscript.

Kind regards,

Azeb Gebresilassie Tesema, Ph.D.

Academic Editor

Journal Requirements:

1. Please provide additional details regarding participant consent. In the ethics statement in the Methods and online submission information, please ensure that you have specified (1) whether consent was informed and (2) what type you obtained (for instance, written or verbal, and if verbal, how it was documented and witnessed). If your study included minors, state whether you obtained consent from parents or guardians. If the need for consent was waived by the ethics committee, please include this information.

2. Please send a completed 'Competing Interests' statement, including any COIs declared by your co-authors. If you have no competing interests to declare, please state "The authors have declared that no competing interests exist". Otherwise please declare all competing interests beginning with the statement "I have read the journal's policy and the authors of this manuscript have the following competing interests:"

3. Please amend your detailed Financial Disclosure statement. This is published with the article. It must therefore be completed in full sentences and contain the exact wording you wish to be published.

1. Please clarify all sources of funding (financial or material support) for your study. List the grants (with grant number) or organizations (with url) that supported your study, including funding received from your institution.

2. State the initials, alongside each funding source, of each author to receive each grant.

3. State what role the funders took in the study. If the funders had no role in your study, please state: “The funders had no role in study design, data collection and analysis, decision to publish, or preparation of the manuscript.”

4. If any authors received a salary from any of your funders, please state which authors and which funders.

5. Your manuscript is missing the following sections: Introduction. Please ensure these are present, and in the correct order, and that any references to subheadings in your main text are correct. An outline of the required sections can be consulted in our submission guidelines here:

https://journals.plos.org/globalpublichealth/s/submission-guidelines#loc-parts-of-a-submission

6. Please provide separate figure files in .tif or .eps format.

7. We have noticed that you have a list of Supporting Information legends in your manuscript. However, there are no corresponding files uploaded to the submission. Please upload them as separate files with the item type 'Supporting Information'.

Reviewers' comments:

Reviewer's Responses to Questions

**Comments to the Author**

1. Does this manuscript meet PLOS Global Public Health’s publication criteria? Is the manuscript technically sound, and do the data support the conclusions? The manuscript must describe methodologically and ethically rigorous research with conclusions that are appropriately drawn based on the data presented.? Is the manuscript technically sound, and do the data support the conclusions? The manuscript must describe methodologically and ethically rigorous research with conclusions that are appropriately drawn based on the data presented.

Reviewer #1: Yes

Reviewer #2: Yes

2. Has the statistical analysis been performed appropriately and rigorously?

Reviewer #1: Yes

Reviewer #2: Yes

3. Have the authors made all data underlying the findings in their manuscript fully available (please refer to the Data Availability Statement at the start of the manuscript PDF file)?

The PLOS Data policy requires authors to make all data underlying the findings described in their manuscript fully available without restriction, with rare exception. The data should be provided as part of the manuscript or its supporting information, or deposited to a public repository. For example, in addition to summary statistics, the data points behind means, medians and variance measures should be available. If there are restrictions on publicly sharing data—e.g. participant privacy or use of data from a third party—those must be specified.requires authors to make all data underlying the findings described in their manuscript fully available without restriction, with rare exception. The data should be provided as part of the manuscript or its supporting information, or deposited to a public repository. For example, in addition to summary statistics, the data points behind means, medians and variance measures should be available. If there are restrictions on publicly sharing data—e.g. participant privacy or use of data from a third party—those must be specified.

Reviewer #1: Yes

Reviewer #2: Yes

4. Is the manuscript presented in an intelligible fashion and written in standard English?

Reviewer #1: Yes

Reviewer #2: Yes

Reviewer #1: Reviewer Comments

1. The criteria for defining “good” competency (≥3.5) appear arbitrary. Please provide justification, references, or validation for this cut-off. How was this threshold selected and why? Response rates for the individual-level surveys are not explicitly stated. Without knowing how many of the invited staff completed the survey, it is difficult to assess representativeness.

2. While the manuscript frequently references the WHO PHEOC framework, there is insufficient discussion of how the findings align with International Health Regulations (IHR 2005) requirements and Joint External Evaluation (JEE) recommendations. Reviewer 1 highlighted this gap, and I strongly encourage the authors to connect observed capacities and gaps more explicitly with JEE indicators in both the results and discussion.

3. Results: The results section is dense and descriptive, and at times blends recommendations into the narrative (e.g., “Ethiopia still needs to improve its data processing…”). The results should focus on observations, with interpretation and recommendations reserved for the discussion. Consider adding a summary table comparing institutional characteristics (legal basis, infrastructure, information systems, training approaches, etc.) across the three countries. This would improve clarity and readability.

4. Discussion: The discussion confirms that PHEOCs were useful during COVID-19, but it does not adequately explore enabling factors such as Nigeria’s prior polio PHEOC experience or lessons from Ebola in West Africa. Including these historical enablers would strengthen the analysis. Sustainability challenges (e.g., funding streams, staff turnover, maintenance of surge capacity) deserve more explicit treatment. As written, many of the “best practices” read more as achievements than sustainable systems.There is little critical reflection on the limits of self-reported data, which likely inflated reported competencies. This limitation should be acknowledged.

Minor Concerns

• Table formatting: In Table 1, some Ethiopia entries are left-aligned and appear under the Senegal column; please correct alignment.

• Acronyms: Ensure all acronyms are spelled out at first mention (e.g., PHEM, JEE, RRTs). Some inconsistencies remain.

• COVID-19 formatting: Hyphenation is inconsistent across the manuscript; please standardize.

• Language: Terms such as “good” and “poor” to describe capacity are overly simplistic. Consider using more precise descriptors (e.g., “adequate,” “limited,” “in need of strengthening”).

• Tables 1 to 3: Table 1 could be strengthened by including training duration and modality (virtual, online, or in-person). Tables 2 and 3 should be clearly labeled as “Self-reported…” to reflect the nature of the data.

• References: Please review all citations to ensure WHO guidance documents and other key references are up to date and correctly formatted.

Reviewer #2: General comments

The paper is an interesting observation of the function and capabilities of PHEOCs in three countries: Nigeria, Senegal and Ethiopia. The study design is solid and the analysis of the data appropriate – however the paper could be improved by digging a bit deeper into providing context for the findings, and drawing out the key lessons learned and recommendations that can be applied in a more general sense. This would elevate the paper from what almost reads like a programmatic evaluation into more robust implementation research.

Specific comments

- Author list: Please check numbering – some authors may have been mistakenly listed as affiliated with WAHO instead of US CDC (currently no authors are listed as affiliated with US CDC and yet US CDC remains in the list below…)

- Line 37: Add “is” between “there” and “limited”.

- Line 38: Minor but consider spelling WHO out in full here.

- Line 41: Add “2020” after “6 December”.

- Line 55: If space/word count allows, adding 1-2 sentences here to briefly describe some of the most important lessons would make the abstract more impactful.

- Line 67: Suggest saying “PHEOCs frequently use…” instead of “PHEOC uses…” – first of all not all PHEOCs use IMS, and also “PHEOC uses” is less grammatically correct.

- Line 70: Remove “the” between “promote” and “best”.

- Line 71: Suggest “In addition” instead of “Besides”.

- Lines 85-86: Should read “there is limited evidence “ (instead of “are”), unless the authors prefer “there are limited data”.

- Line 92: Need to make sure COVID is fully capitalized, not “CoVID”.

- Line 109: The word “data” is plural so it should read “Data were collected…”

- Line 110: Should be “included” not “include”.

- Line 114: Include a citation or at minimum a website for Survey Monkey (and suggest capitalizing it).

- Line 163: Suggest “operations” instead of “operation”

- Table 1: If possible, consider adding citations/links to the materials mentioned, perhaps in the caption?

- Table 2: Use two decimal places throughout for consistency.

- Line 301: Suggest rephrasing to “Legal authority defines a PHEOC’s mandate…”

- Line 302: Maybe a comma is missing after “process”?

- Line 305: “Our results indicate” would sound more natural in English.

- Discussion (in general): How do these findings compare with other papers written about the performance of the PHEOCs in these countries (i.e. https://gh.bmj.com/content/6/10/e007203)? Is it clear that progress has been made? It would be helpful also here to reference to JEE reports from each of the countries to note to what extent the strengths and weakness identified in this study align with what was previously seen through the JEE process, and/or where improvements may have already been made in the meantime (or new capabilities developed through the COVID-19 experience). There needs to be overall a bit more effort to place the study’s findings in context and provide additional analysis, rather than rehashing and summarizing the results.

- Line 365: Should be “Notably”, not “Notable”.

- Conclusions: The paper hints at ways that the findings can provide lessons learned for other countries, but never seems to state them explicitly. Suggest providing some clear lessons learned in the Conclusions section, that directly link to the findings from the study.

**Do you want your identity to be public for this peer review?** For information about this choice, including consent withdrawal, please see our Privacy Policy..

Reviewer #1: No

Reviewer #2: No

---

## [Decision Letter · Decision Letter 1]

16 Mar 2026

Implementing public health emergency operations centres according to an international framework in Ethiopia, Nigeria, and Senegal: Best practices and achievements, 2021

PGPH-D-25-01752R1

Dear Mrs Senait Tekeste Fekadu,

We are pleased to inform you that your manuscript 'Implementing public health emergency operations centres according to an international framework in Ethiopia, Nigeria, and Senegal: Best practices and achievements, 2021' has been provisionally accepted for publication in PLOS Global Public Health.

Best regards,

Azeb Gebresilassie Tesema, Ph.D.

Academic Editor

Reviewer Comments (if any, and for reference):

Reviewer's Responses to Questions

**Comments to the Author**

Reviewer #1: All comments have been addressed

Reviewer #2: All comments have been addressed

publication criteria? Is the manuscript technically sound, and do the data support the conclusions? The manuscript must describe methodologically and ethically rigorous research with conclusions that are appropriately drawn based on the data presented.? Is the manuscript technically sound, and do the data support the conclusions? The manuscript must describe methodologically and ethically rigorous research with conclusions that are appropriately drawn based on the data presented.

Reviewer #1: Yes

Reviewer #2: Yes

3. Has the statistical analysis been performed appropriately and rigorously?

Reviewer #1: N/A

Reviewer #2: Yes

4. Have the authors made all data underlying the findings in their manuscript fully available (please refer to the Data Availability Statement at the start of the manuscript PDF file)?

The PLOS Data policy requires authors to make all data underlying the findings described in their manuscript fully available without restriction, with rare exception. The data should be provided as part of the manuscript or its supporting information, or deposited to a public repository. For example, in addition to summary statistics, the data points behind means, medians and variance measures should be available. If there are restrictions on publicly sharing data—e.g. participant privacy or use of data from a third party—those must be specified.requires authors to make all data underlying the findings described in their manuscript fully available without restriction, with rare exception. The data should be provided as part of the manuscript or its supporting information, or deposited to a public repository. For example, in addition to summary statistics, the data points behind means, medians and variance measures should be available. If there are restrictions on publicly sharing data—e.g. participant privacy or use of data from a third party—those must be specified.

Reviewer #1: Yes

Reviewer #2: Yes

5. Is the manuscript presented in an intelligible fashion and written in standard English?

Reviewer #1: Yes

Reviewer #2: Yes

**Reviewer #1:** My comments were fully addressed by the authors My comments were fully addressed by the authors My comments were fully addressed by the authors My comments were fully addressed by the authors

**Reviewer #2:** Thank you for the detailed consideration of my suggestions. Thank you for the detailed consideration of my suggestions. Thank you for the detailed consideration of my suggestions. Thank you for the detailed consideration of my suggestions.

**Do you want your identity to be public for this peer review?** For information about this choice, including consent withdrawal, please see our Privacy Policy..

Reviewer #1: No

Reviewer #2: No
